

# Degradation of lignocelluloses in straw using AC-1, a thermophilic composite microbial system

Hongdou Liu[1,2], Liqiang Zhang[3], Yu Sun[1], Guangbo Xu[1], Weidong Wang[3], Renzhe Piao[1], Zongjun Cui[4] and Hongyan Zhao[1]

[1] Yanbian University, Yanji, China
[2] College of Land and Environment, Shenyang Agricultural University, Shenyang, China
[3] Heilongjiang Provincial Key Laboratory of Environmental Microbiology and Recycling of Argo-Waste in Cold Region, College of Life Science and Technology, Heilongjiang Bayi Agricultural University, Daqing, China
[4] China Agricultural University, Beijing, China

## ABSTRACT

In composting, the degradation of lignocellulose in straw is problematic due to its complex structures such as lignin. A common solution to this problem is the addition of exogenous inoculants. AC-1, a stable thermophilic microbial composite, was isolated from high temperature compost samples that can decompose lignocellulose at 50–70 °C. AC-1 had a best degradation efficiency of rice straw at 60 °C (78.92%), of hemicellulose, cellulose and lignin were 82.49%, 97.20% and 20.12%, respectively. It showed degrad-ability on both simple (filter paper, absorbent cotton) and complex (rice straw) cellulose materials. It produced acetic and formic acid during decomposition process and the pH had a trend of first downward then upward. High throughput sequencing revealed the main bacterial components of AC-1 were *Tepidimicrobium*, *Haloplasma*, *norank-f-Limnochordaceae*, *Ruminiclostridium* and *Rhodothermus* which provides major theoretical basis for further application of AC-1.

## INTRODUCTION

Urbanization, as well as the steady rise in human population, has resulted in the generation of large quantities of waste globally. These waste streams have led to a number of challenges (environmental, social, and economic) in developing countries (*Goodman, 2020*; *Meena et al., 2018*). Large amounts of straw and livestock manure are produced every year, especially in China. Rice straw is an abundant lignocellulosic waste material in many parts of the world (*Pedraza-Zapata et al., 2017*). For each kg of cropped grain, 1 to 1.5 kg of rice straw is produced. Effective management strategies are needed to overcome the challenges and reduce this large amount of rice straw waste (*Moh & Manaf, 2017*).

Composting is a reliable waste treatment option that could be useful in ameliorating the negative effects that arise when applying organic waste to soil. Composting provides sanitized and stabilized products, which could be utilized as potential sources of organic fertilizers or soil amendments (*Qian et al., 2014*). The thermophilic conditions provide the

Corresponding author
Hongyan Zhao, zhy@ybu.edu.cn

opportunity for total hygienization of compost, by destroying pathogenic organisms present in waste (*Kulikowska, 2016*; *Pandey et al., 2016*). Compared to anaerobic fermentation technology, aerobic fermentation is less expensive, and brings significant economic benefits. The heat can still reach more than 70 °C for days in the northeast region when environmental temperature below 0 °C. This energy produced by large-scale aerobic composting is also one of the available clean energy in cold regions. The composting process results in organic matter decomposition and the formation of humic substances, which can be deployed in the treatment of soil contaminated with heavy metals (*Kulikowska et al., 2015*). Therefore, if used as a bioremediation option, composting can play a significant role as a stabilizer (immobilizes metal in the soil) and as a washing agent, due to the humic substances.

Unfortunately, composting of raw straw is inefficient due to the complex structure of lignocellulose, which is hard to degrade. The addition of exogenous inoculants could shorten composting time while increase composting efficiency (*Shikata et al., 2018*). Bioconversion of lignocelluloses is a better approach by means of microbial co-cultures or complex communities due to synergistic interaction, compared to single bacteria (*Wongwilaiwalin et al., 2010*). Composite microorganism interaction has been proposed as an efficient way to deconstruct biomass (*Jiménez, Korenblum & Elsas, 2014*). To this effect, attempts have been made to characterize co-cultures or specify consortia as starting points for biotechnological processes (*Jiménez, Korenblum & Elsas, 2014*). Due to the obvious advantages, compound inoculants have been gradually recognized and widely used in the composting process (*Zhang et al., 2014*). Few studies have been done on screening of composite strains with high degradation rate while adapt to the extreme environment above 60 °C. In our study, the stable AC-1 isolated from high temperature compost was found have the ability to efficiently degrade lignocellulose especially aerobic, as demonstrated by both screening process and determination of content of lignocellulose components indicators in the degradation process. The optimal decomposition conditions of AC-1 were studied by pH value, temperature and decomposition substrates (filter paper, absorbent cotton, rice straw); DGGE and high-throughput sequencing were used to analyze the bacterial composition of AC-1, so as to provide microbial strains for agricultural waste degradation, contribute to the sustainable development of agriculture.

## MATERIALS & METHODS

### Material
Rice straw was obtained locally after harvest from the experimental fields at YanBian University Jilin, China, cut into pieces, approximately 3–5 cm in length, for further use.

### Screen and train of lignocellulose degrading microbial consortium
PCS medium was composed of 0.5% peptone, 0.1% yeast extract, 0.2% $CaCO_3$, 0.5% NaCl and deionized $H_2O$. The medium was autoclaved at 121 °C for 20 min. PCS medium (75 ml) was added to 100 ml flasks, rice straw was added as 10% (W/V) of the medium volume with a clipped filter paper strip attached to the wall of the flask to observe microbe activity.

Original inoculum sample was taken from the compost with pig manure and rice straw in high temperature period and then inoculated into PCS medium (*Yi et al., 2017*). After inoculation (original inoculum sample, 5% volume of PCS), static intermittent ventilation cultured at 60 °C (each generation has five repetitions), selected the culture with the best decomposing ability (filter paper broken first) and the first generation strains were obtained. Then the liquid of first-generation were inoculated into new fresh medium (5 ml per 100 ml medium) (same as original inoculum sample), and cultured under the same conditions. Cultures with the best decomposing ability were selected, the second-generation strains were obtained. Repeating in the same manner as described above up to the 30th generation, the relatively stable decomposition capacity strains named AC-1 were obtained.

The obtained AC-1 inoculated into new PCS medium for fermentation about 15 days, and samples were taken on day 0 (immediately after inoculation) and on days 1, 3, 5, 7, 9, 12, and 15. Liquid and straw of the culture medium were stored for analysis at −20 °C in centrifuge tubes and sealed bags, respectively.

## Measurements

The pH value was measured using a HORIBA Compact pH meter (Model B-212, Japan). The hemicellulose, cellulose, and lignin content were determined based on van's washing fiber analysis, as described in (*Zhao et al., 2014*) using ANKOM220, USA. The volatile fatty acids (VFA) were determined based on the method (*Peng et al., 2010*; *Wang, Li & Estrada, 2011*) using gas chromatography (GC-7890N, Agilent Inc. USA).

Decomposition ability was measured using weighing. 5% volume of AC-1 was inoculated into each PCS medium. 50 °C, 55 °C, 60 °C, 65 °C, 70 °C were set in the treatments of rice straw as the sole carbon source, each treatment was repeated three times; Filter paper, absorbent cotton and untreated rice straw were set in the same temperature (60 °C) treatments, each treatment was repeated three times, cultured under static intermittent ventilation. After 15 days, the solid substance in each PCS medium was treated as Updegraff's method (*Updegraff, 1969*), dried at 105 °C and weighed.

DNA was extracted using Soil DNA Purification Kit, the bacterial V3–V4 region was amplified by primers 338F (5′-ACTCCTACGGGAGGCAGCAG-3′) and 806R (5′-GGACTACHVGGGTWTCTAAT-3′), PCR reactions were as *Zheng et al. (2020)* described. Gel imager (Alpha Innotech 2200, USA) was used to detected gel electrophoresis (DGGE) products about 15d fermentation samples of AC-1, using 1% agarose under 302 nm ultraviolet (product quantity 5 μl, buffer 1×TAE, electric voltage 100 V, time 30 min, coloration 15 min). Miseq High-throughput sequencing were performed by Majorbio Co, Ltd., China (*Zhang et al., 2015*). Raw reads were deposited in NCBI Sequence Read Archive database and analyzed. All measurements were performed in triplicate.

## Data analysis

Drawing analysis was used Origin 8 mapping software. SPSS 17.0 data analysis software were used for data analysis. Uparse software divides 16S rRNA sequences into operational taxons (OTUs) with 97% threshold.

## RESULTS AND DISCUSSION

### Degradation characteristics of AC-1

As shown in Fig. 1, the degradation trend of rice straw along with its composition of cellulose and hemicellulose showed similar change. On 15 d, the quality of rice straw and the content of hemicellulose, cellulose, lignin decreased by 78.92%, 82.49%, 97.20%, and 20.12%, respectively, in the presence of AC-1. The most readily used components, cellulose and hemicellulose, were rapidly decomposed by microorganisms, and thus, the rice straw was degraded the most intensively during the early fermentation times, but after 9d, they leveled off while the degradation rate of lignin slightly higher in this period. Lignin was found relatively difficult to be directly utilized by microorganisms (*Mei et al., 2020*), thus to improve the degradation rate of lignin is critical. Compared with other bacterial strains (*Guo et al., 2008*; *Seesatat et al., 2020*), the decomposition capacity of AC-1 on each composition is relatively stronger. Studies of degradation capacity by fungi are extensive and deep (*Sanchez, 2009*; *Wang, Han & Liang, 2017*) because of its high efficiency, actually fungi are not suitable for large-scale industrial production by the sensitivity to their surroundings and the temperature. Bacteria are characterized by their variety of sources, pH tolerance, cultivation temperature and their fields of application and in our study, AC-1 are found a stronger efficiency on hemicellulose and cellulose compared with Fungi (*Mishra et al., 2017*; *Mustafa, Poulsen & Sheng, 2016*). Recent years, many researchers are focus on isolating bacteria with ability of degrading lignin including *Bacillus*, *Cyclospora* and *Pseudomonas* (*Aston et al., 2016*) and recombine them for utilization (*Dar et al., 2018*). Different from AC-1 screened by external elimination method through 30 generations, single bacteria and recombinant compound strains are found easy to be replaced by indigenous microorganisms and it's hard to be active in external environment such as composting (*Jiménez, Korenblum & Elsas, 2014*).

Liquid degradation products produced by AC-1 were analyzed. As shown in Fig. 2, the predominant VFAs were acetic and formic acid, along with a few propionic acid (under detected limit) during degradation process. The concentration of formic acid was 0.334 g/L on 1 d, reached a maximum (0.816 g/L) at 9 d, and gradually decreased to a concentration of 0.778 g/L on 15 d. The concentration of acetic acid was 0.519 g/L at 1 d, reached a maximum at 0.85 g/L at 3 d, and decreased to 0.280 g/L on 15 d. The total acid value reached a maximum of 1.37 g/L on 3 d, 1.06 g/L on 15 d, and thus, tended to first increase and then decrease which was basically consistent with the trend of pH value (Fig. 2). Acetic acid was found beneficial for methane fermentation in subsequent application (*Wang et al., 2013*), formic acid is an ideal carrier of hydrogen energy (*Eppinger & Huang, 2017*). These organic acids laying the foundation for development and application of clean energy by straw (*Yuan et al., 2016*).

### Degrad-ability of AC-1

The degradation rate is greatly affected by temperature, as shown in Fig. 3, AC-1 has a certain decomposition effect on rice straw when temperature was from 50 °C to 70 °C. At 50 °C, 55 °C, 60 °C, 65 °C and 70 °C, the decomposition rates were 55.1%, 63.2%,
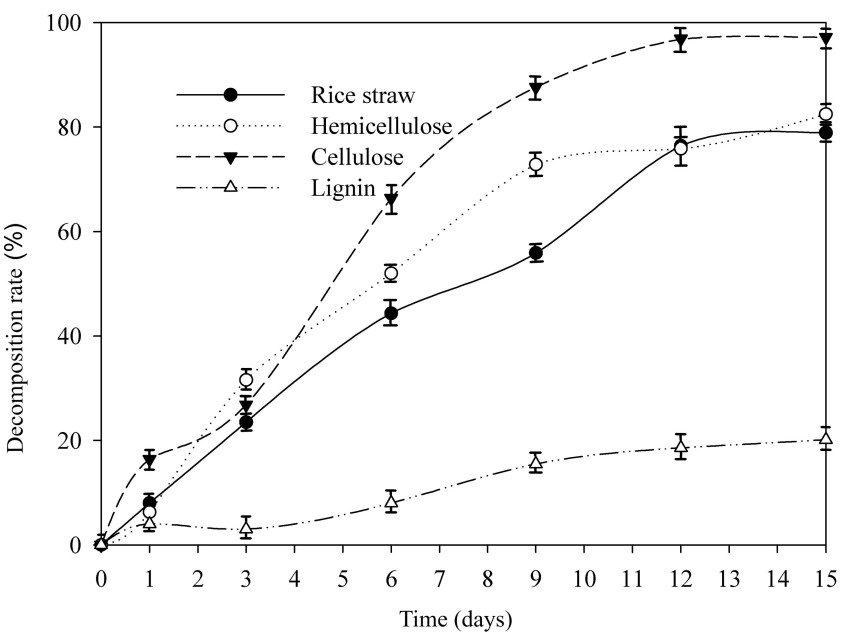

**Figure 1 Changes of cellulose, hemicellulose, and lignin of straw by AC-1 during cultivation.**

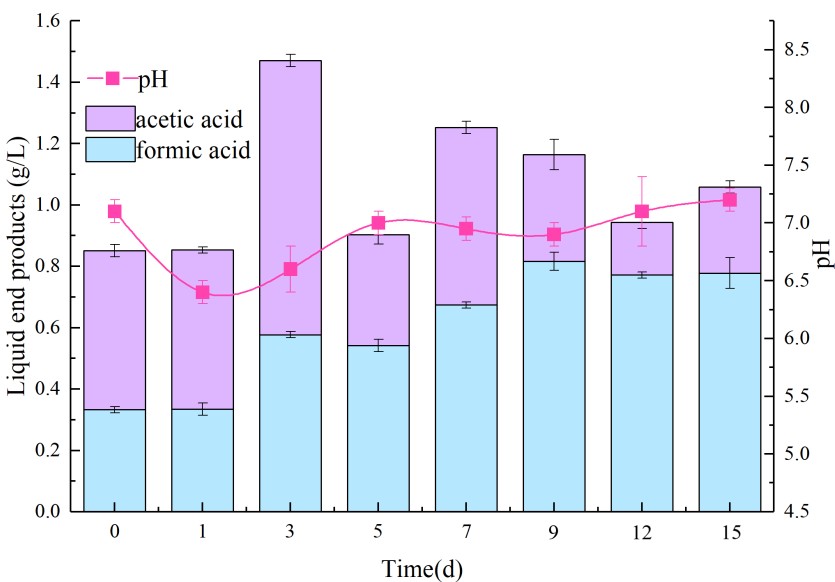

**Figure 2 Changes of volatile organic acids and pH in the degradation process of AC-1 composite strains.**

78.92%, 66.3% and 64.8%, respectively. The best decomposition temperature of AC-1 was about 60 °C (78.92%), which is the common and long-lastingtemperature in thermophilic period of large-scale aerobic composting. Studies show that the thermophilic stage of composting is crucial on the destruction, fracture and decomposition of lignin, cellulose

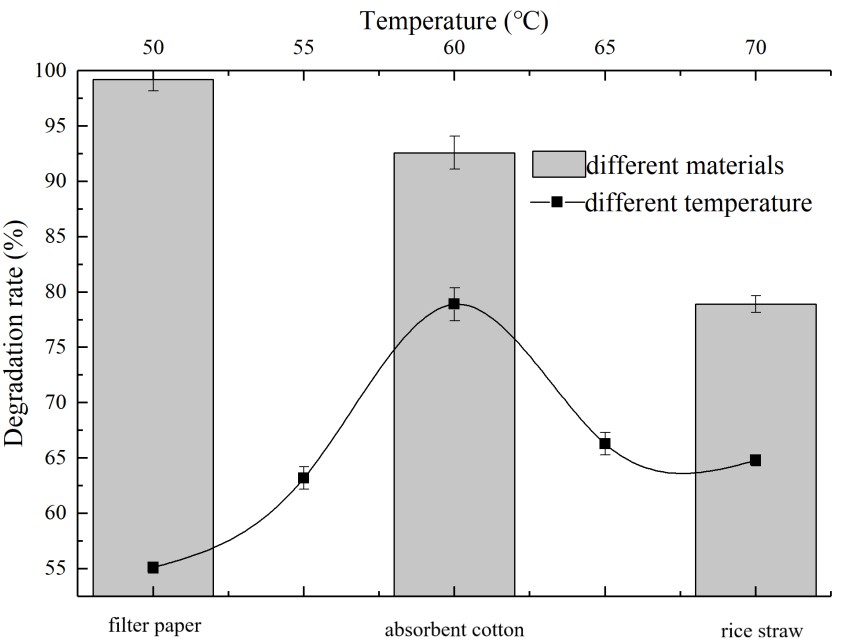

**Figure 3** **Degradation ability of AC-1 under different temperatures and for different materials.**

and hemicellulose, especially over 55 °C (*Hemati et al., 2021*). Therefore, the screened AC-1 could provide a new idea for the further degradation of lignocellulose in the process of large-scale aerobic composting.

AC-1 showed significant difference degradation ability of different materials. The degradation rate of filter paper was the highest, almost all filter paper was decomposed within 5 d, with an average degradation rate of 99.2% after 15 d, followed by absorbent cotton 92.6% and rice straw 78.9% (Fig. 3). It can be seen that AC-1 has strong degradation ability to pure cellulose materials and also to complex cellulose structure.

## Microbial composition

DGGE map can help us find the stable reproduction time of microorganisms. In the whole degradation process, the number of bands with the same brightness were observed after 5d (Fig. 4). Therefore, the stable 12 d AC-1 culture medium was chosen for high-throughput sequencing and a total of 20,630 valid sequences were obtained. At phylum level, there were five main groups (Table 1), among which Firmicutes dominantly accounted for 66.46%, followed Teneriates and Bacteroidetes accounted for 19.11% and 7.16%, respectively, other bacteria accounted for only 7.27% of the total.

The bacteria at the level of class, order, family and genus are also classified, and the percentages are shown in pie charts of Fig. 5. At class level, Clostridia accounted for 41.03%. Mollicutes and Limnochordia accounted for 19.11% and 15.55%, respectively, other classes of bacteria account for about 24.32% of the total (Fig. 5A). At order level, Clostridiales accounted for 37%, Haloplasmatales 19.11%, Limnochordales 15.55%, other orders of bacteria account for about 26.47% (Fig. 5B). At family level, Family-XI-o-clostridiales,

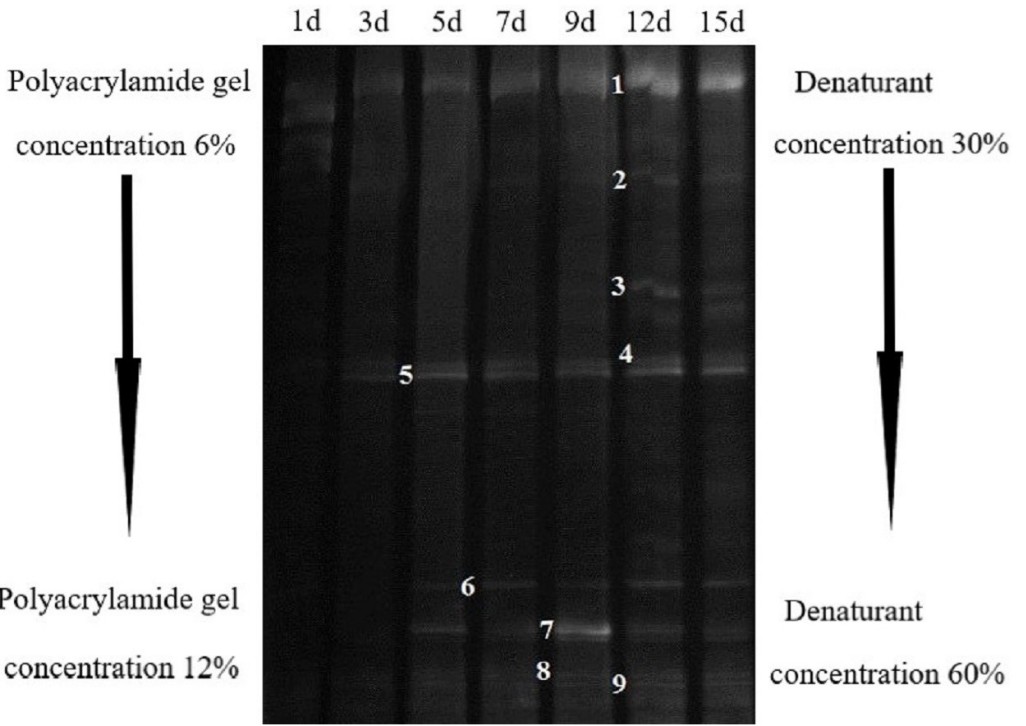

**Figure 4  Bacteria DGGE banding analysis of AC-1 composite strains.**

**Table 1  Bacterial taxonomy at the phylum level.**

| Classification | Relative abundance (%) |
|---|---|
| Firmicutes | 66.46 |
| Teneriates | 19.11 |
| Bacteroidetes | 7.16 |
| Proteobacteria | 6.05 |
| Others | 1.22 |

Haloplasmataceae, Limnochordaceae, Ruminococcaceae and Rhodothermaceae accounted for 20.21%, 19.11%, 13.27%, 11.17% and 7.16%, respectively, other families accounted for about 27.23% (Fig. 5C). Haloplasmataceae is an anaerobic halophilic bacterium, which can reduce the content of nitrate and nitrite (*Antunes et al., 2008*). Limnochordaceae is a moderately thermophilic, facultatively anaerobic, pleomorphic bacterium (*Watanbe, Kojima & Fukui, 2015*). Under Ruminococcaceae family, there are many cellulose decomposing bacteria, mainly produce formic acid, acetic acid and a series of VFA (*Langda et al., 2020*). Rhodothermaceae is thermophilic and halophilic. At genus level, *Tepidimicrobium* accounted for 20.21%, *Haloplasma* 19.11%, *norank-f-Limnochordaceae* 12.94%, *Ruminiclostridium* 9.36%, *Rhodothermus* 7.16%, other genera accounted for about 31.23% (Fig. 5D). *Tepidimicrobium* belongs to Clostridiales, separated from hot springs, can
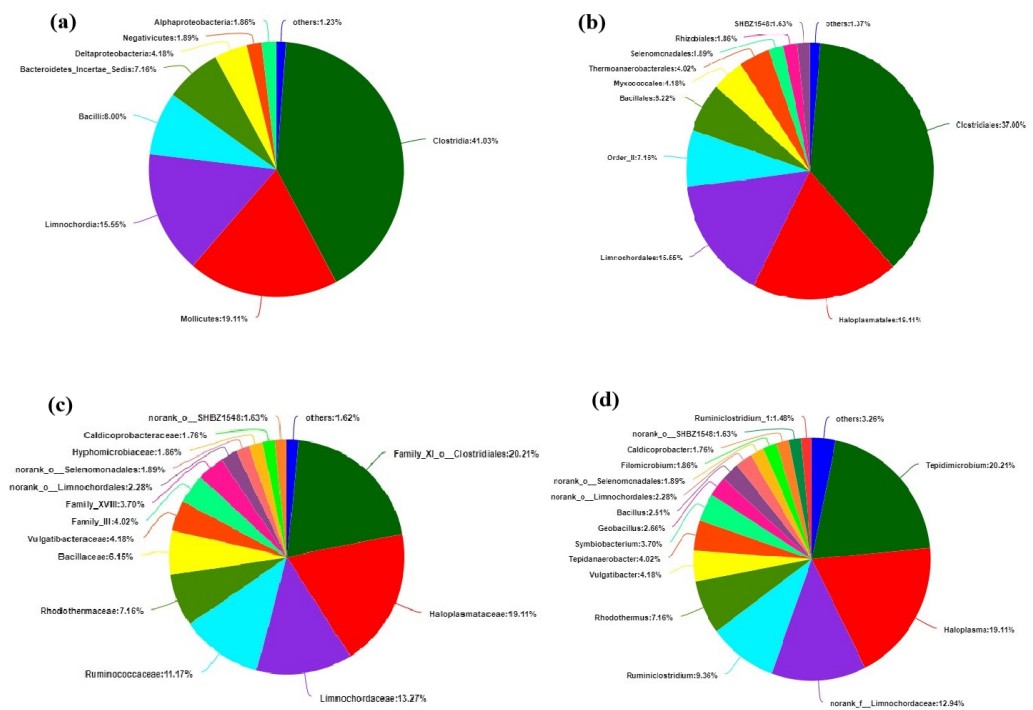

**Figure 5** **Classification of bacteria at different levels.** (A) Class level. (B) Order level. (C) Family level. (D) Genus level.

propagate at 26–62 °C, thermophilic, pH 5.5–9.5 and played a role in cellulose degradation (*Koeck et al., 2013*; *Slobodkin et al., 2020*). *Haloplasma* is known as its deep lineage, vast habitats, salt and acid tolerance (*Surhone, Tennoe & Henssonow, 2011*). *Limnochordaceae* is salt-tolerant and thermophilic (*Watanbe, Kojima & Fukui, 2015*). *Ruminiclostridium* survives in the stomach of ruminants, and help animals decompose cellulose (*Langda et al., 2020*) and *Rhodothermus* is a thermophilic, halophilic bacterium (*Alfredsson et al., 1988*).

# CONCLUSION

A novel active thermophilic lignocelluloses degrading microbial consortium AC-1 has been obtained after 30 generation in this study. It showed best degradation efficiency of rice straw at 60 °C (78.92%) and had degrad-ability on both simple and complex cellulose materials. The acetic and formic acid produced by AC-1 during decomposition process and the pH showed a trend of first downward then upward. The main bacterial components of AC-1 were *Tepidimicrobium*, *Haloplasma*, *norank-f-Limnochordaceae*, *Ruminiclostridium*, *Rhodothermus* by High throughput sequencing. Our study demonstrates that AC-1 can degrade lignocellulose, especially refractory lignin (20.12%) efficiently, provides the theoretical basis and data for the use of AC-1 in accelerating compost maturity and improving quality, efficiency of reuse agricultural wastes.

### Funding

This research was supported by the Special Fund for Agro-scientific Research in the Public Interest (No. 201503137), the Department of Science & Technology of Jilin Province (No. JJKH20191130KJ), the Department of Key R&D Project of Jilin Province Science and Technology (No. 20200402040N C) and supported by Open Foundation of the Heilongjiang Provincial Key Laboratory of Environmental Microbiology and Recycling of Argo-Waste in Cold Region (No. 201711). The funders had no role in study design, data collection and analysis, decision to publish, or preparation of the manuscript.

### Grant Disclosures

The following grant information was disclosed by the authors:
Special Fund for Agro-scientific Research in the Public Interest:  201503137.
Department of Science & Technology of Jilin Province:  JJKH20191130KJ.
Department of Key R&D Project of Jilin Province Science and Technology:  No. 20200402040N C.
Open Foundation of the Heilongjiang Provincial Key Laboratory of Environmental Microbiology and Recycling of Argo-Waste in Cold Region:  201711.

### Competing Interests

The authors declare there are no competing interests.

### Author Contributions

- Hongdou Liu conceived and designed the experiments, performed the experiments, analyzed the data, prepared figures and/or tables, authored or reviewed drafts of the paper, and approved the final draft.
- Liqiang Zhang and Yu Sun performed the experiments, prepared figures and/or tables, and approved the final draft.
- Guangbo Xu, Renzhe Piao and Zongjun Cui conceived and designed the experiments, authored or reviewed drafts of the paper, and approved the final draft.
- Weidong Wang analyzed the data, authored or reviewed drafts of the paper, and approved the final draft.
- Hongyan Zhao conceived and designed the experiments, prepared figures and/or tables, and approved the final draft.

### DNA Deposition

The following information was supplied regarding the deposition of DNA sequences:
The sequences are available at NCBI: PRJNA744399, SAMN20089760, SRR15241520, SRX11547404.

### Data Availability

The raw measurements are available in the Supplementary Files.

## Supplemental Information

Supplemental information for this article can be found online at http://dx.doi.org/10.7717/peerj.12364#supplemental-information.

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
