# Peer review of "Degradation of lignocelluloses in straw using AC-1, a thermophilic composite microbial system"

_PeerJ, doi:10.7717/peerj.12364_

## Round 0.1 · original submission · Minor Revisions

Two experts in the field reviewed your manuscript and raised some concerns that need to be addressed.

Reviewer 1 ·

Basic reporting

This article introduces the degradation of lignocelluloses in straw using AC-1, a thermophilic composite microbial system. This article introduces the exogenous inoculants. AC-1 on the degradation efficiency of rice straw at different temperature and materials, which provides major theoretical basis for further application of AC-1.
It is interesting study for degrading straw based on exogenous thermophilic composite microbes. However, some details need attention.

Experimental design

Abstract:
1. Line 15-16: the degradation of lignocellulose in straw is problematic, what is the problem? Please state clearly.
2. The experimental design is unclear and the logic is confusing.
3. what is simple and complex cellulose materials?
Introduction:
4. Line 49-50: Please revise, “complex mechanism of lignocellulose” changed to “complex structure of lignocellulose”
Materials & Methods
5. The rice straw is pretreated by NaOH method, please explain why. Are the subsequent series processed with NaoH?
6. Line 79-81: “filter paper broken” refer to enzyme activity or which medium? What is the carbon source?
7. Screening strains and culture degradation experiments are not clear, please revise carefully.
8. Is DGGE at different temperatures or different carbon sources?

Validity of the findings

NONE

Additional comments

Results and discussion
9. Line 118-119: “the content of rice straw” refer to what?
10. Figure 3 is inconsistent with the material method introduced
11. Line 133-134: Genus level should be in italics.
12. The article did not introduce those a thermophilic composite microbial system that are related to temperature or at are related to carbon sources, which can be further explored. The dominant microorganisms at different temperatures or different carbon sources should be further studied. Please discuss the changes in microorganisms in detail. There is no CK treatment in the article.

·

Basic reporting

The idea of using AC-1 for lignocellulose degradation is necessary because the substrate is resistant to degradation. Because the research site is in a cold climate and is not conducive to the growth of microorganisms necessary for the decomposition of rice straw, the researcher must show references from journals in a reliable database that AC-1 is cultured and capable of degrading lignocellulose at temperatures near 0 °C.

Experimental design

The experimental objectives should be consistent with the experimental scope. The researcher had to demonstrate that, in this investigation, AC-1 could actually decompose lignocellulose from rice straw. The process of preparing AC-1 from pig manure (lines 78-86) should be reliably attributed to either the method itself developed or a derivative of any researcher's research (Show references).
The information from the introductory section, the researcher wanted to decompose AC-1 with rice straw, which contains the chemical composition of lignocellulose (lignin bonded cellulose), but the experimental design process was seemed likely that the researcher takes AC-1 to decompose commercial compounds of lignocellulose. Researchers should describe whether they wish to directly decompose lignocellulose from rice straw or to experiment with the above constituents for future applications.

Validity of the findings

From the experimental design and the results, the investigator used the GC for analysis which measured the volatile organic acids and assumed that the experimental results were converted to a reduced total lignocellulose content. In this case, the investogators should explain in the experimental design part and show references.

---

## Round 0.2 · accepted · Accept

The authors have significantly modified the manuscript, following the reviewers' comments.